# How Does Physical Exercise Improve the Subjective Well-Being of the Chinese Adult Population? A Moderated Mediation Model

**DOI:** 10.3390/healthcare12101048

**Published:** 2024-05-20

**Authors:** Zhan Liu, Yufeng Zhao, Tianning Zhang, Dianxi Wang

**Affiliations:** 1School of Social Work, Beijing College of Social Administration, Beijing 102600, China; liuzhan@bcsa.edu.cn; 2Institute of Social Development, Chinese Academy of Macroeconomic Research, Beijing 100038, China; 20210001@ucass.edu.cn; 3School of Marxism, Beijing Sport University, Beijing 100084, China; 13669369232@163.com

**Keywords:** physical exercise, subjective well-being, social capital, Internet use, conditional process analysis

## Abstract

Based on the 2017 data from the China General Social Survey (CGSS), the conditional process analysis method was used to explore the relationship between physical exercise and subjective well-being, as well as the mediating effect of social capital and the moderating effect of Internet use. This study found that the effect of physical exercise on subjective well-being includes both direct effects and indirect effects. Physical exercise directly and positively affects SWB and partially affects SWB through social networks as a mediating variable. Additionally, Internet use can moderate the direct effect of physical exercise on SWB and the mediating effect of social networks on the relationship between the two. As the frequency of Internet use increases, the link between physical exercise and social capital weakens, resulting in a decreased promotion effect on SWB. The results of this study further reveal the internal mechanism linking physical exercise and the improvement of SWB, which is of great significance for enhancing the accessibility of physical exercise facilities and promoting a healthy online lifestyle, ultimately leading to improved well-being among residents.

## 1. Introduction

Subjective well-being (SWB) refers to an individual’s personal evaluation of how closely their current life aligns with their ideal vision, resulting in positive emotions and attitudes [1]. Enhancing the well-being of residents has consistently been a primary goal for governments and international organizations. This pursuit involves various actions and initiatives, such as reducing poverty, mitigating inequality, improving infrastructure, enhancing the environment, and strengthening public health services [2,3,4,5,6]. All of these efforts are geared towards improving the overall welfare of members of society. In China, the sustained and rapid economic growth in the post-reform era has markedly improved the living standards of people. According to the United Nations’ World Happiness Report, China’s happiness ranking has seen a significant increase, moving from 112th in 2012 to 64th in 2022 among 156 nations and regions [7,8]. Previous studies emphasize the role of physical exercise as a crucial lifestyle factor in sustaining and enhancing SWB. These studies assert that physical exercise can improve residents’ SWB by cultivating positive emotions, enhancing social capital such as social networks and social trust, and enabling individuals to maintain a higher level of SWB [9,10,11]; however, the mechanism of social capital in the relationship between physical exercise and SWB has not been fully understood. This study utilizes data from the 2017 Chinese General Social Survey (CGSS) and employs conditional process analysis (CPA) to investigate the impact of social capital and Internet use on SWB.

## 2. Theoretical Background

### 2.1. Physical Exercise and Subjective Well-Being

The correlation between physical exercise and individual SWB has long been the focus of scholars. Previous studies suggest a significant correlation between actively engaging in physical exercise and individual health, well-being, subjective happiness, and overall quality of life [12,13]. Evidence also suggests that physical exercise significantly promotes residents’ SWB [12,14,15,16], with more intensive engagement correlating with higher levels of happiness [17]. Scholars have also found that the frequency and duration of exercise among sports club members have a positive predictive effect on their SWB [11]. In addition, participating in physical exercise not only promotes residents’ SWB but also has a significant positive effect on improving residents’ physical and psychological health [15].

The mechanisms that influence SWB can be categorized into three paradigms: social psychology, behaviorism, and social networks. The social-psychological paradigm focuses on the role of emotional factors in generating well-being; for instance, physical exercise can lead to emotional rewards such as flow, euphoria, and contentment [10], which in turn foster personal satisfaction and happiness. Studies based on the self-determination theory model suggest that the emotional experience derived from exercise significantly contributes to SWB [18]. Additionally, research based on the theory of self-efficacy has shown that one’s sense of self-efficacy predicts engagement in physical exercise. Engaging in physical exercise with clear goals and intentions has been found to enhance self-efficacy, thus fostering SWB [19]. Goal orientation theory also suggests that achieving goals can lead to SWB in individuals. It determines a person’s sense of happiness and is also a primary source of obtaining and maintaining happiness [20,21]. During the process of physical exercise, achieving personal goals can cultivate a positive sense of confidence, foster satisfaction, and nurture positive emotions, ultimately enhancing SWB. Consequently, physical exercise serves as a significant avenue for meeting individual emotional needs, empowering individuals to develop confidence, garner recognition, and receive support, thereby achieving a sense of happiness.

From a behaviorist’s perspective, individuals’ health and happiness are influenced by their lifestyles and habits. Regular physical exercise is associated with better health and greater happiness, while unhealthy behaviors, such as excessive drinking and sedentary habits, are more likely to cause poor health [22]. The representative health behavior theory model categorizes the factors affecting life satisfaction into four dimensions: environment, individual characteristics, health behaviors, and effects. This theoretical model places a particular emphasis on the behavior patterns that affect well-being outcomes.

The social network analysis paradigm places particular emphasis on the happiness derived from social interactions and engagement with others during physical exercise [9]. Some scholars argue that exercise is perceived as a social activity that not only enhances physical health but also nurtures the formation of social networks [23,24]. In this perspective, sports are acknowledged as a tool for tackling social issues and promoting positive social capital, such as improved social cohesion, trust, and interpersonal relationships. The establishment of social capital through sports associations represents a common strategy for strengthening social cohesion. This process fosters mutual connections and builds trust in others, ultimately enhancing social cohesion and reducing social exclusion [25]. Therefore, physical exercise not only satisfies individual health and emotional needs but also expands interpersonal relationships and social networks, thereby promoting social unity.

### 2.2. Social Capital and Subjective Well-Being

Social capital is a resource that individuals access as members of a network or group, derived through social connections [26]. According to social capital theory, as a social resource embedded in interpersonal networks and mobilized to achieve behavioral goals [27], social capital provides mechanisms for connectivity, network structures, and relational resources. This can explain the generative process of social participation [28], individual status attainment [29], health, and well-being [30,31]. Recently, extensive and growing studies have explored the relationship between social capital and SWB. The consistent findings suggest that social capital significantly and positively affects residents’ SWB and quality of life [32,33]. This indicates that richer social capital leads to higher SWB and quality of life. In addition, among different social capital components, social trust notably affects residents’ SWB, while social networks, neighborhood interactions, and friendships also play significant positive roles [33].

Social capital in various contexts also significantly predicts residents’ SWB. For example, scholars have found that community social capital has the most substantial effect on the SWB of elderly rural residents, followed by individual social capital, and then family social capital [32]. Community capital also positively affects community life satisfaction. Residents benefit from strong informal community support networks and formal community engagement resources, leading to higher community life satisfaction [34]. Additionally, family social capital indirectly enhances women’s SWB by alleviating psychological burdens and reducing depression levels, while work-related social capital compensates for the happiness loss caused by reduced income among women by providing a conducive working environment [35]. Family social capital continues to have a significant impact on the life satisfaction of older individuals; however, when controlling the influence of family social capital, cognitive social capital, specific social trust, and general social trust also play a significant role in shaping the life satisfaction of older adults [36]. Furthermore, other scholars have also found that, in the Chinese context, the proportion of friends in New Year greeting networks, the frequency of neighborhood interactions, and informal social participation have a positive effect on life satisfaction, indicating that the expansion of social network size, the increase in the proportion of friends, and the frequency of informal social participation all contribute to the improvement of individual life satisfaction [37]. Based on these findings, this study integrates the concept of social capital into the analysis of the relationship between physical exercise and SWB, aiming to elucidate the underlying mechanisms through which social capital enhances residents’ overall happiness.

### 2.3. Internet Use and Subjective Well-Being

Over the past decades, the rapid development and widespread adoption of the Internet have exerted a substantial influence on both the economic landscape and social life of China. The Internet has become intertwined with various aspects of people’s lives, significantly influencing their SWB [38]. The 49th Statistical Report on the Development of the Internet in China shows that, by December 2021, the number of Internet users in China had reached 1.032 billion, and that the Internet penetration rate had reached 73%. With the increasing influence of the Internet on individuals’ daily lives, the utilization of online resources may also have a potential impact on residents’ SWB.

There are two conflicting perspectives on the relationship between Internet use and individual SWB. The first highlights the potential negative impacts of Internet use on people’s subjective happiness. For instance, the theory of the relationship between media use and depression argues that constant exposure to negative news on the Internet may intensify residents’ dissatisfaction with society, resulting in a decline in social trust and a subsequent decrease in life satisfaction [39]. The “time displacement hypothesis” suggests that spending excessive time online reduces opportunities for social interaction, weakening individuals’ motivation to engage in real-world activities. This can potentially increase feelings of loneliness and detachment, thus lowering overall happiness levels [40,41]. Some empirical studies support these claims; for example, Schiffrin et al. [42] found that increased Internet use frequency is associated with decreased SWB. Excessive reliance on the Internet may also diminish offline emotional connections and reduce enthusiasm for engaging in social activities [43].

The second viewpoint emphasizes how Internet use can enhance people’s SWB. According to the information mobilization theory, the Internet provides individuals with enough autonomy to choose information and enables quick and efficient dissemination, thus lowering the cost of accessing information [44]. For example, scholars have noted that the development of the Internet has enhanced operational efficiency, saved time, and enriched daily life [45], all contributing to increased levels of happiness. Moreover, the openness, anonymity, virtual nature, and equality of interactions on the Internet promote widespread social participation among residents, fostering social and identity recognition, as well as consequently boosting happiness [46].

### 2.4. The Current Study

With the rapid development of the information society, the Internet is fundamentally altering how people consume information, live, and interact socially. Its influence on individuals’ SWB is undeniable. Concurrently, the fusion of the Internet with sports, known as “Internet + Sports”, is reshaping how people engage in physical exercise and sports, with individuals increasingly relying on Internet technology for their fitness needs. In this context, Internet use may enhance residents’ SWB. Building on this premise, this study, guided by the perspective that Internet use fosters SWB, hypothesizes that the frequency of Internet use may enhance the positive effect of physical exercise on happiness by reinforcing the connection between physical exercise and social capital. Therefore, this study introduces Internet use as a moderating variable to elucidate the impact mechanisms through which physical exercise contributes to subjective well-being.

Based on the analysis above, this study has developed a theoretical framework, as depicted in Figure 1, by integrating the principles of social network analysis and theoretical debates on the connections between social capital, Internet usage, and subjective well-being. We assume that participation in physical exercise can not only directly enhance residents’ SWB, but also affect it through the facilitation of social capital. This effect is moderated by Internet use. Specifically, since many physical exercise programs are team-building activities, people may become more familiar with a community or make new friends when they participate in physical exercise, and people could also enhance their trust in others and the collective by exercising together in this process, to expand their social network capital. In addition, the use of the Internet may diminish the accessibility and convenience of people’s face-to-face social interactions, thus weakening the positive correlation between physical exercise and well-being.

### 2.5. Hypotheses

Based on the analytical framework presented in Figure 1, social capital acts as a mediator in the relationship between physical exercise and SWB, while Internet use moderates the association between physical exercise and SWB. Consequently, the following research hypotheses are proposed:

**Hypothesis** **1.**
*Physical exercise positively predicts SWB.*


**Hypothesis** **2.**
*Physical exercise may indirectly affect SWB through the mediating role of social capital.*


**Hypothesis** **3.**
*Internet use moderates the relationships among physical exercise and SWB, physical exercise and social capital, as well as social capital and SWB.*


## 3. Materials and Methods

### 3.1. Data

This study utilizes data from the 2017 China General Social Survey (CGSS). The CGSS, hosted by the National Survey Research Center at Renmin University of China, is a large-scale social survey project that is conducted annually. Since 2003, the CGSS has sampled more than 10,000 households across the country. It is the oldest nationwide, comprehensive, and continuous academic survey project in China. In particular, CGSS 2017 addresses the issue of residents’ Internet usage, providing rare and nationally representative data on individual Internet use in China. CGSS 2017 utilizes a multi-stage stratified sampling method, selecting respondents from 29 provinces, municipalities, and autonomous regions throughout China. Face-to-face interviews were conducted with respondents, totaling 12,582 samples. Since Module D of the questionnaire to which the SWB scale belongs was designed separately from the 2016 East Asian General Social Survey, only 4132 respondents were surveyed. Therefore, cases with missing data on the SWB variable were excluded, and only observations with complete information were retained. The final sample size was 4060. The average age of the valid sample was 50.893 years, with the youngest being 18 years and the oldest being 96 years. The demographic characteristics of the sample are presented in Table 1.

### 3.2. Variables

#### 3.2.1. Dependent Variable

The dependent variable in this study is subjective well-being. CGSS 2017 uses the abbreviated version of the Chinese Subjective Well-being Scale (SWBS-CC20) developed by Xing [47] to assess SWB. This scale encompasses 10 dimensions with 20 items, covering experiences related to adaptation to interpersonal relations, mental health, goals and personal values, psychological balance, physical health, family atmosphere, confidence towards society, growth and progress, satisfaction and abundance, and self-acceptance. Each item is rated on a 6-point scale, ranging from “strongly disagree”, “disagree”, “somewhat disagree”, “somewhat agree”, to “strongly agree”, with scores ranging from 1 to 6, respectively. In this study, reverse items are converted, and the overall SWB score is derived by summing the values of each item. The minimum value is 20, with a maximum of 120. The Cronbach’s α of this scale is 0.852.

#### 3.2.2. Independent Variable

The independent variable in this study is physical exercise, which is measured by a single question from the questionnaire: “Over the past 12 months, how many times per week, on average, do you engage in physical exercise that lasts at least 30 min and makes you sweat?” This variable is continuous, ranging from 0 to 70, with a mean value of 2.323.

#### 3.2.3. Mediating Variables

The definition of social capital lacks universal consensus, and existing research exhibits diverse interpretations as well as measurement approaches; however, it commonly encompasses dimensions such as social trust and social networks. In this study, these two indicators are adopted to measure social capital based on the existing literature. Following the study by Li and Chen [48], two questions, “Over the past year, how frequently do you engage in social and entertainment activities with your neighbors?” and “Over the past year, how often do you participate in social and entertainment activities with other friends?”, were chosen to measure social networks. The responses to these questions included “Never” (1), “Once a year or less” (2), “A few times a year” (3), “About once a month” (4), “A few times a month” (5), “Once or twice a week” (6), and “Almost every day” (7). The variable of social networks is derived by summing the values of these two indicators. Drawing from the measurement approach outlined by Zhang and Wan (2020) [6], the question “In general, do you believe that, in this society, most people can be trusted?” is utilized to measure social trust. Responses to this question include five options: “Strongly disagree” (1), “Disagree” (2), “Neither agree nor disagree” (3), “Agree” (4), and “Strongly agree” (5).

#### 3.2.4. Moderating Variable

The moderating variable examined in this study is Internet use. It is measured through the question “Over the past year, how frequently have you used the internet (including mobile internet)?”. The response options were “Never” (1), “Rarely” (2), “Occasionally” (3), “Frequently” (4), and “Very frequently” (5).

#### 3.2.5. Control Variables

Previous studies have found that SWB exhibits significant individual differences and varies according to gender, age, ethnicity, and socioeconomic status [49,50]; therefore, control variables such as gender, age, ethnicity, hukou status, education level, marital status, and political affiliation are included in our analysis. Gender is a binary variable, with 1 representing male and 2 representing female. Age is a continuous variable ranging from 18 to 96. Ethnicity is categorical, with 1 denoting a minority ethnic group and 2 denoting Han ethnicity. Hukou is categorized into agricultural hukou (1) and non-agricultural hukou (2). Education level is a continuous variable ranging from 0 to 18. Marital status includes unmarried (1) and married (2). Political affiliation comprises non-party members (1) and party members (2).

### 3.3. Statistical Methods

In this study, the basic profile of the sample was described using descriptive statistical analysis. Independent sample t-tests and F-tests were conducted to compare the mean differences of demographic variables related to the key variables. Additionally, the study utilizes the conditional process analysis method proposed by Hayes [51] to examine the mediating role of social capital and the moderating effect of Internet use. Model 4 (simple mediation model) from Hayes’ SPSS macro is used to estimate the mediating effect of social capital on the relationship between physical exercise and SWB. The bootstrap method was also applied to estimate a 95% confidence interval based on 2000 random samples to test the significance of the mediation effect. If the 95% confidence interval does not contain 0, it is statistically significant. Subsequently, Model 59 from the SPSS macro is applied to test the moderated mediation model, considering relevant control variables. The formulated model equations are as follows:(1)$$Y=\lambda_{0}+\lambda_{1}X+\lambda_{2}C_{1}$$
(2)$$M=\beta_{0}+\beta_{1}X+\beta_{2}M_{i}+\beta_{3}C_{2}$$
(3)$$Y=\gamma_{0}+\gamma_{1}X+\gamma_{2}M_{i}+\gamma_{3}W+\gamma_{4}XW+\gamma_{5}M_{i}W+\gamma_{6}C_{3}$$

In the given equations, *X*, *M*, *W*, and *Y* correspond to the independent variable, mediator variable, moderator variable, and dependent variable, respectively. *C*_1_, *C*_2_, and *C*_3_ represent control variables. To examine how variable *W* moderates the mediating effect (conditional process), the regression equation incorporates the main effects of the moderator variable *W*, along with interaction terms between *W* and *X*, as well as *W* and *M*. Furthermore, the parameter tests for *γ*_4_ and *γ*_5_ help ascertain whether *W* significantly moderates the direct effect of the independent variable, *X*, on the dependent variable, *Y*, and the effect of the mediator variable, *M*, on *Y*. The moderated mediation model requires a simple slope analysis, which involves substituting values for the moderator variable at low levels (*W* mean minus 1 standard deviation, W − 1SD), moderate levels (*W* mean), and high levels (*W* mean plus 1 standard deviation, *W* + 1SD) into the equation. This approach illuminates the variations in direct, indirect, and total effects under different values of the moderator variable.

## 4. Results

### 4.1. Descriptive Analysis Results

Table 2 displays differences in the SWB of respondents according to various demographic and social characteristics. Males exhibit significantly higher SWB than females. Han ethnicity respondents tend to report higher SWB compared to their ethnic minority counterparts, but the difference is not statistically significant. The SWB of non-agricultural residents is significantly higher than that of agricultural residents. Respondents who are members of the Communist Party show a significant increase in SWB compared to non-members. Additionally, respondents with spouses demonstrate significantly higher SWB than those without spouses. The descriptive statistical analysis highlights that SWB among respondents varies based on gender, hukou status, political affiliation, and marital status; however, according to the results of Cohen’s *d* and Cohen’s *f*, respondents with different hukou and political statuses have more significant differences in SWB.

### 4.2. Conditional Process Analysis Results 

In this study, Model 4 from Hayes’ SPSS macro plugin PROCESS v3.3 was used to examine the mediating effects of social capital (social networks and social trust) on the relationship between physical exercise and SWB. This analysis controlled for variables such as gender, age, ethnicity, hukou, political affiliation, educational level, and marital status, as presented in Table 3 and Table 4. In Table 3, Model 1 revealed a significant positive effect of physical exercise on SWB. At a significance level of 0.001, each unit increase in physical exercise corresponds to a 0.211-unit increase in residents’ SWB scores, supporting hypothesis one of this study. Further examination in Models 2 and 3 indicates that physical exercise significantly affects social networks but not social trust. In Model 4, after including the mediating variable of social capital, the direct predictive effect of physical exercise on SWB remained significant. Controlling for mediating and demographic variables, each unit increase in physical exercise is associated with a 0.189-unit increase in residents’ SWB scores. 

According to Table 4, the upper and lower limits of the bootstrap 95% confidence interval for both the direct effect of physical exercise on SWB and the mediating effect of social networks do not contain 0. This indicates that physical exercise not only directly affects SWB, but also influences it through the mediating role of social networks. The direct effect of physical exercise on SWB is 0.189, while the mediation effect of social networks is 0.016. These account for 89.57% and 7.58% of the total effect, respectively. However, the mediating effect of social trust is not significant, offering partial support for hypothesis two.

This study employs Model 59 from Process v3.3, which assumes that the various paths of the mediating model are moderated, aligning with the theoretical framework of this study. After controlling for variables such as gender, age, ethnicity, hukou, political affiliation, education level, and marital status, the moderated mediation model is examined, as presented in Table 5 and Table 6. As shown in Model 6 of Table 5, Internet use demonstrates a positive effect on SWB. This suggests that for each unit increase in Internet use frequency, the social network score increases by 0.104. The interaction term between physical exercise and Internet use on social networks is significant. This suggests that as Internet use increases, the impact of physical exercise on social networks decreases by 0.018. Based on Model 7 in Table 5, an increase in the frequency of Internet use can enhance SWB. Specifically, the SWB score of residents increases by 0.317 for every unit increase in the frequency of Internet use. After including the moderating variable of Internet use in Model 7, the interaction term between physical exercise and Internet use on SWB remains significant. Moreover, as the frequency of Internet use increases, the impact of physical exercise on SWB decreases by 0.05; however, the effects of physical exercise on social networks and social trust are not significant. These findings indicate that Internet use not only moderates the direct effect of physical exercise on SWB but also moderates the predictive effect of physical exercise on social networks.

This study also conducted a simple slope analysis to illustrate the moderating effect of Internet use, as depicted in Figure 2 and Figure 3. Figure 2 shows that the social networks of residents who frequently use the Internet are larger than those of residents who use the Internet less frequently, regardless of whether their physical exercise frequency is high or low. For respondents with a lower frequency of Internet use (W − 1SD), physical exercise significantly and positively predicts social networks (simple slope = 0.085, *t* = 4.998, *p* < 0.001); however, for individuals with a higher frequency of Internet use (W + 1SD), the impact of physical exercise on social networks is not statistically significant (simple slope = 0.023, *t* = 1.322, *p* = 0.186). This result suggests that Internet use acts as a moderator in the relationship between physical exercise and social networks. Specifically, as the frequency of Internet use increases, the impact of physical exercise on social networks decreases.

Figure 3 shows that residents with a high frequency of Internet use report higher levels of SWB compared to those with a low frequency of Internet use, regardless of their engagement in high- or low-frequency physical exercise. For respondents with a low Internet use frequency (W − 1SD), physical exercise significantly and positively predicts SWB (simple slope = 0.262, *t* = 4.595, *p* < 0.001). Conversely, for individuals with a high frequency of Internet use (W + 1SD), the impact of physical exercise on SWB is not statistically significant (simple slope = 0.088, *t* = 1.504, *p* = 0.132). This suggests that Internet use moderates the relationship between physical exercise and SWB. As Internet use frequency increases, the predictive effect of physical exercise on SWB decreases, especially among individuals with high Internet use frequency, where physical exercise does not affect SWB.

Table 6 presents the bootstrap results of a moderated mediation effect model. Based on Table 6, the mediating effect of social capital on the correlation between physical exercise and SWB varies across three levels of Internet use. Among respondents with low and medium Internet use frequencies, the mediating effect of social networks on the relationship between physical exercise and SWB is significant. Conversely, for those with high frequencies of Internet use, this mediating effect is not significant. Furthermore, the mediating effect of social trust is also not significant. These findings indicate that for individuals with medium and low Internet use frequencies, the mediating role of social networks in the relationship between physical exercise and SWB is more significant. As Internet use frequency increases, physical exercise is less likely to enhance SWB by expanding social networks. In summary, both the direct impact of physical exercise on SWB and the mediating influence of social networks are moderated by Internet use, partially supporting the third hypothesis of this study.

## 5. Discussion

This study, based on the data from CGSS 2017, examines the mediating effect of social capital on the relationship between physical exercise and SWB, as well as the moderating role of Internet use in this association. We found that, firstly, even after controlling for variables such as gender, age, ethnicity, hukou, political affiliation, education level, and marital status, physical exercise has a significant positive effect on SWB. Secondly, social networks play a mediating role in the relationship between physical exercise and SWB. Physical exercise can enhance individual SWB by expanding social networks; however, the mediating effect of social trust is not significant. Third, Internet use moderates the direct effect of physical exercise on SWB, as well as the mediating role of social networks.

The findings of this study are in line with research on the correlation between physical exercise and SWB. For instance, participating in physical exercise can significantly predict an individual’s SWB [12,15,16,17]. The mediating role of social networks in the relationship between physical exercise and SWB also aligns with the findings of studies conducted by Liu [10] and Kim [52]. In the process of participating in physical exercise, individuals may establish more familiar contacts with their communities, neighbors, and even strangers. In this process, individuals can enhance their trust in others and the collective by exercising together and experiencing positive emotions, such as identification, hope, and optimism, to enhance their SWB. Moreover, the increase in Internet use frequency is correlated with individual SWB, which also supports the relevant view that Internet use can improve SWB [46]. This study found that, as the frequency of Internet use increases, physical exercise is less likely to enhance SWB by expanding social networks. Excessive or long-term use of the Internet cannot help people find opportunities or resources to participate in physical exercise, but may reduce the time that people have available to participate in physical exercise, thus reducing the accessibility and frequency of social interaction, further decreasing people’s SWB and weakening the correlation between physical exercise and SWB. This highlights the influence of Internet use on the connection between physical exercise and SWB. 

This study has certain limitations: Regarding the measurement of social capital, in addition to the methods utilized in this study, there are various measurement methods in the existing literature. For example, Spring Festival greeting networks can be used as a measurement of social networks, or the division of social capital into structural social capital and cognitive social capital can be used for operational measurement. Based on previous studies and the structure of the CGSS questionnaire, this study adopts a measurement approach commonly used in the relevant literature; however, incorporating multiple measurement methods for mutual validation could enhance the accuracy of the findings. Furthermore, due to data limitations, this study only used a single item to measure Internet use instead of a multidimensional scale, which may cause problems in terms of internal consistency.

## 6. Conclusions

Drawing upon the aforementioned analysis, this study emphasizes that, on the one hand, social capital acts as a mediating factor between physical exercise and SWB. Strengthening the connection between physical exercise and social capital can enhance the positive impact of physical exercise on SWB. On the other hand, as Internet use frequency increases, the connection between physical exercise and social capital weakens, leading to a hindering effect on SWB. In other words, excessive or prolonged Internet use may not necessarily lead to an increase in social networks or foster an elevation in SWB. The main contribution of this study lies in providing further insights into the mechanisms linking physical exercise and SWB. It delineates the influence of physical exercise on individual SWB, as well as the roles of social capital and Internet use in this relationship, thereby enriching the research on the association between physical exercise and SWB.

The findings of this study have significant policy implications. Firstly, the government must enhance the development of grassroots community sports facilities. Drawing from the actual fitness needs of communities, efforts should be intensified to increase the construction and renovation of community sports facilities. This aims to provide the necessary infrastructure for various demographics to engage in physical activities in their community, continually improving the accessibility and convenience of such facilities. Secondly, when the government aims to achieve universal Internet access and bridge the “digital divide”, there should be a concerted emphasis on promoting the “Internet + Sports + Happiness” initiative. Encouraging a positive and healthy online lifestyle can significantly contribute to enhancing residents’ overall well-being and happiness. Thirdly, it is essential to establish a strong network of grassroots community sports organizations. This involves the creation of local sports associations and the promotion of diverse sports community initiatives. By fostering community interaction networks and enriching social capital, organizations provide support to facilitate community sports participation and enhance happiness.

## Figures and Tables

**Figure 1 healthcare-12-01048-f001:**
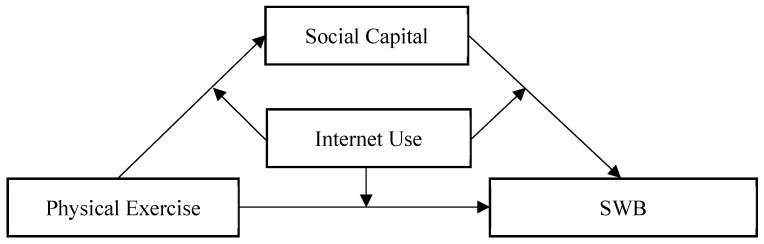
Theoretical model depicting the relationships among physical exercise, social capital, internet use, and SWB.

**Figure 2 healthcare-12-01048-f002:**
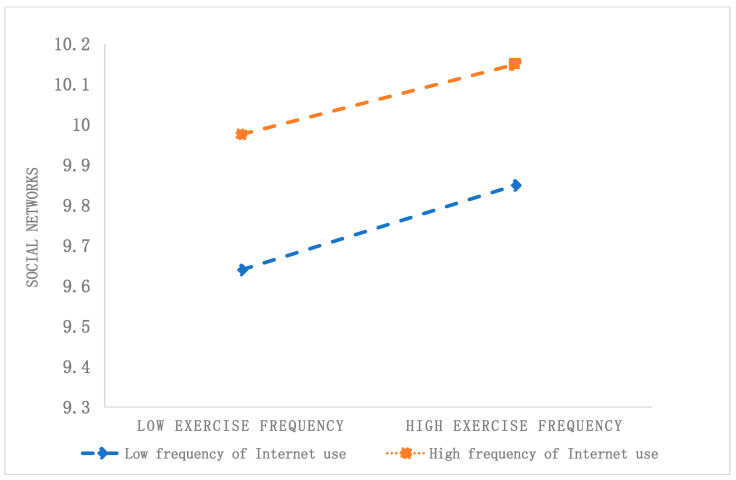
The moderating effect of Internet use on the relationship between physical exercise and social networks.

**Figure 3 healthcare-12-01048-f003:**
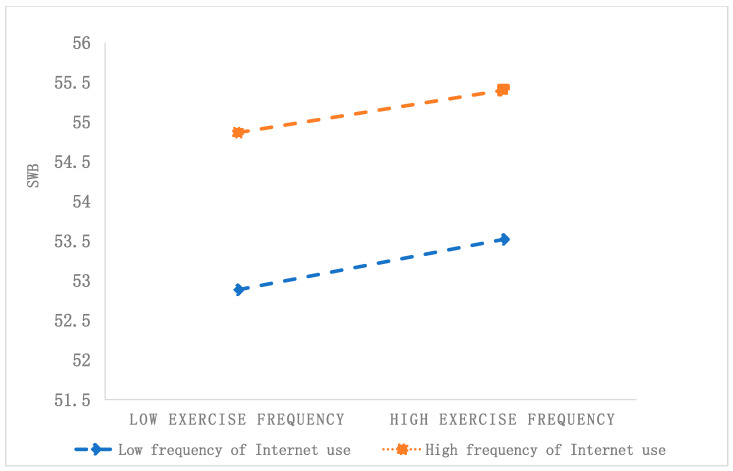
The moderating effect of Internet use on the relationship between physical exercise and SWB.

**Table 1 healthcare-12-01048-t001:** Sample distribution (N = 4060).

Characteristics	Frequency	Percentage (%)
Gender		
Male	1880	46.31
Female	2180	53.69
Ethnicity		
Ethnic minority	297	7.32
Han nationality	3763	92.68
Hukou Type		
Agricultural hukou	2125	52.42
Non-agricultural hukou	1929	47.58
Political Affiliation		
Non-party member	3401	83.77
Party member	659	16.23
Marital Status		
Unmarried	968	23.84
Married	3092	76.16
Age Groups		
10–35	891	21.95
36–45	614	15.12
46–60	1245	30.67
Over 60	1310	32.27

**Table 2 healthcare-12-01048-t002:** SWB among respondents with various characteristics.

Characteristics	Mean ± Standard Deviation	Test Value	Significance	Effect Size
Gender				
Male	82.527 ± 12.603	*T* = 2.676	*p* = 0.008	Cohen’s *d* = 0.084
Female	81.452 ± 12.887			
Ethnicity				
Ethnic minority	81.013 ± 12.806	*T* = −1.313	*p* = 0.189	Cohen’s *d* = −0.079
Han nationality	82.024 ± 12.761			
Hukou				
Agricultural hukou	79.599 ± 12.443	*T* = −12.543	*p* < 0.001	Cohen’s *d* = −0.394
Non-agricultural hukou	84.536 ± 12.618			
Political Affiliation				
Non-party member	80.962 ± 12.692	*T* = −11.372	*p* < 0.001	Cohen’s *d* = −0.494
Party member	87.046 ± 11.911			
Marital Status				
Unmarried	80.458 ± 14.074	*T* = −4.176	*p* < 0.001	Cohen’s *d* = −0.148
Married	82.417 ± 12.293			
Age Groups				
10–35	84.792 ± 11.288	*F* = 29.96	*p* < 0.001	Cohen’s *f* = 0.149
36–45	83.673 ± 11.261			
46–60	80.878 ± 13.017			
Over 60	80.227 ± 13.685			

**Table 3 healthcare-12-01048-t003:** Results of the mediation effects model.

	Model 1: SWB			Model 2: Social Trust			Model 3: Social Networks			Model 4: SWB		
	Coefficient	Standard Error	t-Value	Coefficient	Standard Error	t-Value	Coefficient	Standard Error	t-Value	Coefficient	Standard Error	t-Value
Physical exercise	0.211 ***	0.040	5.261	0.004	0.003	1.022	0.057 ***	0.011	4.897	0.189 ***	0.039	4.770
Social trust										1.700 ***	0.183	9.289
Social networks										0.272 ***	0.052	5.168
Gender (reference group: male)	−0.011	0.385	−0.030	−0.004	0.032	−0.117	0.272 *	0.113	2.398	−0.079	0.381	−0.208
Age	−0.017	0.013	−1.260	0.010 ***	0.001	8.672	−0.015 ***	0.004	−3.727	−0.030 *	0.013	−2.216
Hukou (reference group: agricultural hukou)	1.667 ***	0.455	3.661	−0.075 *	0.038	−1.965	−0.580 ***	0.134	−4.326	1.954 ***	0.450	4.337
Education level	0.631 ***	0.059	10.592	0.005	0.005	0.991	−0.003	0.017	−0.183	0.623 ***	0.058	10.605
Marital status (reference group: unmarried)	2.824 ***	0.449	6.289	0.048	0.038	1.261	0.148	0.132	1.126	2.701 ***	0.443	6.097
Ethnicity (reference group: ethnic minority)	−0.136	0.730	−0.186	−0.107	0.061	−1.736	0.031	0.214	0.147	0.037	0.720	0.052
Political affiliation (reference group: non-party member)	3.008 ***	0.555	5.418	0.156 ***	0.047	3.323	0.132	0.163	0.813	2.706 ***	0.548	4.933
Constant	66.442 ***	2.040	32.570	2.952 ***	0.172	17.085	10.701 ***	0.601	17.820	58.505 ***	2.158	27.110
R-sq	0.114			0.027			0.017			0.138		
F	64.940 ***			14.113 ***			8.865 ***			64.727 ***		

Note: * *p* < 0.05; and *** *p* < 0.001.

**Table 4 healthcare-12-01048-t004:** Decomposition of total, direct, and mediating effects.

	Effect Value	Boot Standard Error	Boot CI Lower Bound	Boot CI Lower Bound	Relative Effect Proportion
Total effect	0.211	0.049	0.131	0.327	-
Direct effect	0.189	0.047	0.117	0.303	89.57%
Mediating effect on social networks	0.016	0.005	0.007	0.029	7.58%
Mediating effect on social trust	0.006	0.006	−0.005	0.018	2.84%

**Table 5 healthcare-12-01048-t005:** Results of moderated mediation model.

	Model 5: Social Trust			Model 6: Social Networks			Model 7: SWB		
	Coefficient	Standard Error	t-Value	Coefficient	Standard Error	t-Value	Coefficient	Standard Error	t-Value
Physical exercise	0.003	0.006	0.490	0.104 ***	0.023	4.453	0.317 ***	0.078	4.027
Social trust							1.938 ***	0.352	5.497
Social networks:							0.298 **	0.093	3.195
Internet use	−0.014	0.015	−0.933	0.317 ***	0.052	6.085	1.931 ***	0.508	3.799
Physical exercise × Internet use	0.0001	0.002	0.047	−0.018 *	0.007	−2.447	−0.050 *	0.024	−2.057
Physical exercise × social networks							−0.073	0.105	−0.700
Physical exercise × social trust							−0.026	0.031	−0.848
Gender (reference group: male)	−0.003	0.032	−0.104	0.271 *	0.113	2.396	−0.092	0.378	−0.243
Age	0.009 ***	0.001	6.891	−0.001	0.004	−0.180	0.034 *	0.015	2.161
Hukou (reference group: agricultural hukou)	−0.069	0.039	−1.757	−0.731 ***	0.135	−5.387	1.249 **	0.455	2.741
Educational level (teference group: primary school and below)	0.007	0.005	1.253	−0.035 *	0.018	−1.927	0.468 ***	0.061	7.570
Marital status (reference group: unmarried)	0.049	0.038	1.304	0.110	0.131	0.837	2.555 ***	0.440	5.800
Ethnicity (reference group: ethnic minority)	−0.105	0.061	−1.704	−0.009	0.214	−0.044	−0.154	0.715	−0.216
Political affiliation (reference group: non-party member)	0.156 ***	0.047	3.328	0.131	0.162	0.806	2.704 ***	0.544	4.967
Constant	2.995 ***	0.178	16.759	9.744 ***	0.618	15.758	53.199 ***	2.569	20.704
R-sq	0.027			0.026			0.152		
F	11.383 ***			10.869 ***			51.712 ***		

Note: * *p* < 0.05; ** *p* < 0.01; and *** *p* < 0.001.

**Table 6 healthcare-12-01048-t006:** Mediating effects across different levels of Internet use.

	Internet Use	Effect Value	Boot Standard Error	Boot CI Lower Bound	Boot CI Lower Bound
Mediating effect on social trust	W − 1SD (1.106)	0.006	0.010	−0.012	0.029
	W (2.823)	0.006	0.006	−0.005	0.020
	W + 1SD (4.540)	0.006	0.008	−0.010	0.022
Mediating effect on social networks	W − 1SD (1.106)	0.023	0.011	0.008	0.049
	W (2.823)	0.012	0.005	0.005	0.025
	W + 1SD (4.540)	0.004	0.004	−0.002	0.014

## Data Availability

Data are contained within the article.

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
