# Peer review of "How Does Physical Exercise Improve the Subjective Well-Being of the Chinese Adult Population? A Moderated Mediation Model"

_healthcare, 2024, doi:10.3390/healthcare12101048_

Round 1

Reviewer 1 Report

Comments and Suggestions for Authors

Review for manuscript healthcare-2981985

Physical Exercise, Internet Use, and Subjective Well-being: A Moderated Mediation Model

Thank you for presenting your manuscript, describing an empirical study with 4,060 participants, drawn from the China General Social Survey. The authors test a model (Figure 1) that should explain the relationship between exercise and subjective well-being by introducing social capital and internet use in addition. In the following I will present some comments and suggestions for improvement.

Page 1ff. Please check your manuscript for any typos and syntax errors. For example, on p. 1 I found three typos, (l. 30: enhancing, l. 34: improved, l. 39: enabling), l. 98 includes a sentence with an unknown mistake in the syntax. In line 129 you are mentioning an inverted U relationship between life satisfaction and social network, but later you describe a positive correlation between three variables and life satisfaction. This would mean that the empirical situation should like like “the more the better” which contradicts and inverted U. Please check that paragraph in greater detail.

On p. 4, last paragraph, you describe your model in the following way: Physical exercise may enhance residents’ SWB and facilitate social capital. This effect should be moderated by internet use. Figure 1 is supposed to mirror that model. The way it is described the arrows in the model should represent positive or negative associations and these could be detected by structural equation models. Please clarify why you preferred the statistical method that you used.

On p. 5, you describe the sample. Please clarify why the sample decreased from more than 12000 participants to 4,060 which represents less than one third of the original sample (32.3 percent). In line 215f. you give the reason of missing data. Does this mean that two third gave no data on your variables? How did this  sample reduction influence the representativeness of your data? Can you give information which groups of your sample may have denied to answer more or less and how may this have influenced the results? Please give also information on the age of the participants, as this variable may also have influenced your results.

In your data assessment (p. 5, 233-260) you have most variables assessed with kind of Likert scales from 1-5, 1-6, 1-7. Respondents had to give estimations. One variable (physical activity) is estimated for the past year very exactly (how many times per week) and can reach a number from 0 to 70. 70 in this definition would mean that on average, a participant could do exercise 70 times per week? Did you check for outliers? The mean in your sample is 2.3 (l. 238). Whereas physical activity is estimated for the whole past year, social networks and social trust are estimated for the current situation (how frequently do you engage in activities (1-7 and 1-5 respectively). But internet use is again measured for the past year and participants estimate the frequency from 1 to 5.

Please understand that I am a little bit suspicious about the validity of your data, as they obviously cover different time ranges and may be influenced by memory distortions.

Results: Please present d-values in Table 2. As your sample is big, p alone is very distorting. And why did you not check for age differences. All you variables may be different depending on the age of your participants.

Pearson correlations in Table 3 are quite low and reach significance due to the high number of participants. But they never reach more than 10% determination coefficient.

As I told formerly, the different time spans of your variables make it difficult to make a meaningful analysis and interpretation of your data. Do you think that current social relations may influence former physical activity or vice versa former physical activity may influence current social relations? I would be happy to hear your comments. 

Comments on the Quality of English Language

All in all, I feel that English is appropriate. Please look at my first comment and check your manuscript for typos and syntax errors.

Author Response

Response to Reviewer 1:

Thank you for your suggestions on this manuscript, and I have revised it according to your comments.

(1) Page 1ff. Please check your manuscript for any typos and syntax errors. For example, on p. 1 I found three typos, (l. 30: enhancing, l. 34: improved, l. 39: enabling), l. 98 includes a sentence with an unknown mistake in the syntax. In line 129 you are mentioning an inverted U relationship between life satisfaction and social network, but later you describe a positive correlation between three variables and life satisfaction. This would mean that the empirical situation should like like “the more the better” which contradicts and inverted U. Please check that paragraph in greater detail. 

Response: Thank you for your comment. I have revised it according to your suggestions. Meantime, I checked my entire manuscript for any typos and syntax errors.

(2) On p. 4, last paragraph, you describe your model in the following way: Physical exercise may enhance residents’ SWB and facilitate social capital. This effect should be moderated by internet use. Figure 1 is supposed to mirror that model. The way it is described the arrows in the model should represent positive or negative associations and these could be detected by structural equation models. Please clarify why you preferred the statistical method that you used.

Response: Thank you for your comment. I agree that the structural equation model can also test the relationship between physical exercise, internet use and SWB. However, we use the Process approach primarily for the following reasons. First of all, since only four core variables are involved in this study, the test of the path relationship between these four core variables can be analyzed with the Process approach, and the structural equation model method may be more suitable for testing the relationship between more variables. Meantime, the Process approach can also allow us to specify of who is the independent variable and who is the moderating variable. Second, the four core variables mentioned above do not involve latent variables, so the Process method proposed by Hayes may be more appropriate for this study. Third, structural equation models are not convenient for simple slope tests, including the drawing of effect graphs. To sum up, the Process method proposed by Hayes is adopted in this study.

(3) On p. 5, you describe the sample. Please clarify why the sample decreased from more than 12000 participants to 4,060 which represents less than one third of the original sample (32.3 percent). In line 215f. you give the reason of missing data. Does this mean that two third gave no data on your variables? How did this sample reduction influence the representativeness of your data? Can you give information which groups of your sample may have denied to answer more or less and how may this have influenced the results? Please give also information on the age of the participants, as this variable may also have influenced your results.

Response: Thank you for your comment. Since Module D of the CGSS 2017 questionnaire to which the SWB scale belongs was designed separately with the 2016 East Asian General Social Survey, and only 4132 respondents were surveyed. Therefore, cases with missing data on the SWB variable were excluded and only observations with complete information were retained. The final sample size was 4060 people, and this leads to many missing data. (See page 5; lines 306-309.)

Regarding the age of the respondents, I have added the following content. That is, “The average age of the valid sample was 50.893 years, with the youngest being 18 years and the oldest being 96 years.” (See page 5; lines 310-311.)

(4) In your data assessment (p. 5, 233-260) you have most variables assessed with kind of Likert scales from 1-5, 1-6, 1-7. Respondents had to give estimations. One variable (physical activity) is estimated for the past year very exactly (how many times per week) and can reach a number from 0 to 70. 70 in this definition would mean that on average, a participant could do exercise 70 times per week? Did you check for outliers? The mean in your sample is 2.3 (l. 238). Whereas physical activity is estimated for the whole past year, social networks and social trust are estimated for the current situation (how frequently do you engage in activities (1-7 and 1-5 respectively). But internet use is again measured for the past year and participants estimate the frequency from 1 to 5. Please understand that I am a little bit suspicious about the validity of your data, as they obviously cover different time ranges and may be influenced by memory distortions. 

Response: Thank you for your comment. First of all, about outliers, we checked the data again and found that the sample size of more than 40 sports exercises is relatively large, and it is difficult for us to judge whether these are singular values. In reality, there are indeed many elderly people who like sports, because the elderly in China, especially the elderly in cities, like morning exercise and square dancing. So, it's hard to tell if these samples are invalid singular values. For this reason, we still retained the respondents who answered physical exercise 70 times.

Secondly, as for the validity of your data, we reviewed the questionnaire again. We found that both of the social network measurement questions asked respondents about their social interactions in the past year. That is, for the two questions of "How frequently do you engage in social and entertainment activities with your neighbors?" and "How often do you participate in social and entertainment activities with other friends?", respondents were asked how much they had socialized with friends and neighbors in the past year.

In addition, for social trust, the questionnaire does not indicate the trust attitude of the respondents in the past year. However, we believe that in the absence of dramatic social changes in a short period, people's sense of trust may be relatively stable. For this reason, it may also be acceptable to select this question about social trust from the questionnaire.

(5) Results: Please present d-values in Table 2. As your sample is big, p alone is very distorting. And why did you not check for age differences. All your variables may be different depending on the age of your participants. 

Response: Thank you for your comment. In this revision, I present Cohen’s d values in Table 2. At the same time, we also increased the differences in SWB among respondents of different age groups. (See Table 2 in pages 8)

(6) Pearson correlations in Table 3 are quite low and reach significance due to the high number of participants. But they never reach more than 10% determination coefficient.

Response: Thank you for your comment. In this revision, I deleted the part of Pearson correlation analysis.

(7) As I told formerly, the different time spans of your variables make it difficult to make a meaningful analysis and interpretation of your data. Do you think that current social relations may influence former physical activity or vice versa former physical activity may influence current social relations? I would be happy to hear your comments.

Response: Thank you for your comment. We reviewed the questionnaire again. We found that both of the social network measurement questions asked respondents about their social interactions in the past year. That is, for the two questions, "How frequently do you engage in social and entertainment activities with your neighbors?" and "How often do you participate in social and entertainment activities with other friends?", respondents were asked how much they had socialized with friends and neighbors in the past year.

Reviewer 2 Report

Comments and Suggestions for Authors

Kindly find the comments attached.

Author Response

Response to Reviewer 2:

Dear reviewer,

Thank you for your suggestions on this manuscript, and I have revised it according to your comments.

(1) Provide explanations in what way PE can improve SWB? Describe more - which aspects that have 'yet to be fully understood?

Response: Thank you for your comment. I have revised it according to your suggestions. That is, Previous studies emphasize the role of physical activity as a crucial lifestyle factor in sustaining and enhancing SWB. These studies assert that physical exercise can improve residents’ SWB by cultivating positive emotions, enhancing social capital such as social networks and social trust, and enabling individuals to maintain a higher level of perceived happiness (Frey & Stutzer, 2010; Liu, 2016; Xia et al., 2018; Tan et al., 2020). However, the mechanism of social capital in the relationship between physical exercise and SWB has not been fully understood.” (See page 1; lines 37-40.)

(2) I feel that there is a need to explain more regarding the PE - Social Capital path. Perhaps by re-viewing studies that support this relationship. Also, there is a need to explain how internet use can enhance/reduce the strength of these relationships.

Response: Thank you for your comment. I have revised it according to your suggestions. That is, “Specifically, since many physical exercise programs are team-building activities, people may become more familiar with the community or make new friends when they participate in physical exercise, and people can also enhance their trust in others and the collective by exercising together in this process, to expand their social network capital. In addition, the use of the Internet may diminish the accessibility and convenience of people's face-to-face social interactions, thus weakening the positive correlation between physical exercise and well-being.” (See page 4; lines 259-265.) In addition, in the parts of 1.2 and 1.3, by reviewing previous studies, we discussed the relationship between PE and social capital and the role of Internet use. (See page 2-4; parts 1.2 and 1.3.)

(3) There are many aspects in the construct of internet use that can be measured. In addition, there are also many existing instruments that have been developed to measure internet use. So why do authors still use a single-item measure that is known to have issues related to internal consistency in this study?

Response: Thank you for your comment. I have revised it according to your suggestions. Indeed, there are other scales for measuring Internet use, and the CGSS 2017 questionnaire provides a similar scale in the “Online Society” survey section. However, in the process of data processing, we found that respondents who filled out the “Subjective Well-Being” survey were not asked to fill out the “Online Society” survey section, that is, respondents who answered the subjective well-being scale did not answer the Internet use scale. As a result, we have to use a single item to measure Internet usage. This may be a shortcoming of this study.

(4) There is a huge difference in terms of number between these groups. As such, one might speculate that the significant difference may be attributed to the sample size rather than actual difference.

Response: Thank you for your comment. I have revised it according to your suggestions. It is true that groups may indeed vary significantly due to sample size. Therefore, in this revision, I add d-values in Table 2 to further present the effect size of group differences. (See Table 2, in page 8).

(5) Are these limits showed in Table 4?

Response: Thank you for your comment. The upper and lower limits of the bootstrap 95% confidence interval are displayed in Table 4 and Table 6. Table 4 and Table 6 only present the result of Bootstrap, while Table 3 and Table 5 present the result of multiple regression analysis without involving Bootstrap confidence intervals.

(6) Any bootstrapping analysis conducted in table 7?

Response: Thank you for your comment. Yes. We performed bootstrapping analysis and Table 6 presents the bootstrap results of a moderated mediation effect model.

(7) Maybe better to separate the paragraphs into (1) discussion, and (2) conclusion.

Response: Thank you for your comment. I have revised it according to your suggestions. We divided the part of “Discussion” into two parts: “Discussion” and “Conclusion”.

(8) Why? Discuss why physical exercise affects SWB by social networks?

Response: Thank you for your comment. I have revised it according to your suggestions. That is, “The findings of this study are in line with research on the correlation between physical exercise and SWB. For instance, participating in physical exercise can significantly predict an individual's SWB [12, 15-17]. The mediating role of social networks in the relationship between physical exercise and SWB also aligns with the findings of studies conducted by Liu [10] and Kim [53]. In the process of participating in physical exercise, individuals may establish more familiar contacts with the community, neighbors, and even strangers. In this process, individuals can enhance their trust in others and the collective by exercising together, and experiencing positive emotions such as identification, hope, and optimism, to enhance their SWB. Moreover, the increase in internet use frequency is correlated with individual SWB, which also supports the relevant view that internet use can improve SWB [46]. This study found that as the frequency of Internet use increases, physical exercise is less likely to enhance SWB by expanding social networks. Excessive or long-term use of the Internet cannot help people find opportunities or resources to participate in physical exercise but may reduce people's time to participate in physical exercise, thus reducing the accessibility and frequency of social interaction, and further decreasing people's SWB and weakening the correlation between physical exercise and SWB. This highlights the influence of Internet use on the connection between physical exercise and SWB.” (See page 13; lines 585-602).

(9) So, why don't the authors use the Spring Festival here in this study? Why the authors do not examine the effect in this present study? Why the author do not examine the effect in this present study?

Response: Thank you for your comment. The reason why the Spring Festival was not adopted in this study is mainly related to the questionnaire design of CGSS 2017. The questionnaire of CGSS2017 does not provide survey questions related to the Chinese New Year Greeting Networks. Therefore, it is possible to use the Spring Festival to measure social capital.

Round 2

Reviewer 1 Report

Comments and Suggestions for Authors

Dear author, thank you for your thorough revision of the manuscript.